# Towards Generalizable Personalized Federated Learning with Adaptive Local Adaptation

## Abstract

*Personalized federated learning* aims to find a shared global model that can be adapted to meet personal needs on each individual device. Starting from such a shared initial model, devices should be able to easily adapt to their local dataset to obtain personalized models. However, we find that existing works cannot generalize well on non-iid scenarios with different heterogeneity degrees of the underlying data distribution among devices. Thus, it is challenging for these methods to train a suitable global model to effectively induce high-quality personalized models without changing learning objectives. In this paper, we point out that this issue can be addressed by balancing information flow from the initial model and training dataset to the local adaptation. We then prove a theorem referred to as the *adaptive trade-off theorem*, showing adaptive local adaptation is equivalent to optimizing such information flow based on the information theory. With these theoretical insights, we propose a new framework called *adaptive federated meta-learning* (AFML), designed to achieve generalizable personalized federated learning that maintains solid performance under non-IID data scenarios with different degrees of diversity among devices. We test AFML in an extensive set of these non-IID data scenarios, with both CIFAR-100 and Shakespeare datasets. Experimental results demonstrate that AFML can maintain the highest personalized accuracy compared to alternative leading frameworks, yet with a minimal number of communication rounds and local updates needed.

## 1 Introduction

In recent years, research interests in training a machine learning model on edge devices motivated the paradigm of federated learning (FL) Li et al. (2020), which makes it feasible for multiple devices to collaboratively train a shared global model in a privacy-preserving manner. Recent works Jiang et al. (2019); Fallah et al. (2020); Mansour et al. (2020) explored *personalized federated learning* (PFL) Kulkarni et al. (2020), which aims to produce high-quality personalized models based on device-specific datasets and objectives. Specifically, each device uses the global model shared by the central server for its initial local model. Then, in the local adaptation Yu et al. (2020), it trains a personalized model by performing several local updates with respect to their own data.

On the one hand, Jiang et al. (2019); Arivazhagan et al. (2019) trained a global model to obtain the best accuracy on the whole data distribution of all participants. Thus, such global model with high optimality can be used as a strong start initialization point in personalization for each client with a subset data distribution. On the other hand, the works Fallah et al. (2020); Khodak et al. (2019); Chen et al. (2019); Jiang et al. (2019) tended to train a global model with high adaptability that current or new devices can easily adapt to their datasets in their local adaptation. These approaches are expected to produce high-quality personalized models given the heterogeneity of the underlying data distribution for all the users, i.e., non independent and identically distributed (non-iid) data.

However, we observe that existing frameworks lack the capability to adapt or generalize well between various non-iid data scenarios with different degrees of data and objectives diversity among devices. The learning efficiency and personalized performance of existing approaches are not perfectly performed to the changes in such diversity of non-iid data scenarios. With quantitative experimental

results, we clearly point out that this is caused by targeting an unsuitable trade-off between the optimality and adaptability of the global model, which is referred to as the inappropriate optimality-adaptability trade-off.

The most recent works Hanzely & Richtárik (2020); Mansour et al. (2020); Deng et al. (2020) potentially alleviated this issue by mixing the global model and personalized local models in one objective function for joint optimization. However, these works lack a clear insight of generalized PFL and optimality-adaptability trade-off. In addition, the proposed methods relied upon the combination formula with mixing weights that are hard to update in different non-IID data scenarios.

In this paper, we rethink the local adaptation primarily from an information theory perspective Yin et al. (2020). We argue that the information required to learn a personalized model $\phi$ in local adaptation is derived from both the initial model $\theta$ and the local dataset $D$. Thus, high $I(\phi; \theta)$ and $I(\phi; D)$ correspond to the global model trained to be high optimality and adaptability on the whole data distribution, respectively. This leads us to identify suitable trade-offs in different on-IID data scenarios by balancing two information flow sources. Therefore, with the idea of avoiding the local update from solely relying on one information source, we formalize the learning process with an optimization problem by adding mutual information constraints. The key insight of this approach is that it can maintain performance by adaptively adjusting two information flows in the local adaptation based on the non-IID data condition.

Our mathematical analysis of this optimization problem further gives a lower bound that consists of the meta-learning training loss Finn et al. (2017) and three regularizers. Our proposed *adaptive trade-off theorem* merges these principles to an embodiment. It theoretically ensures that our major contribution in this paper is to achieve generalizable PFL with adaptive local adaptation.

These theoretical insights lead to our design of a novel *adaptive federated meta-learning* (AFML) framework. The basic learning architecture of AFML naturally inherits the meta-learning PFL framework that has been tested by fed-MAML and proven to converge Fallah et al. (2020). Then, the corresponding training algorithm is proposed based on deduced regularization terms of the theorem. We tested AFML on CIFAR-100 and Shakespeare datasets. Our experimental results on non-iid scenarios with extensive degrees of diversity show that AFML maintains the highest personalized accuracy. Moreover, AFML achieves a reduction in required communication rounds by $25.5\%$ and $11.6\%$, and a reduction in communication cost by 1.6 and 3.9 times compared with alternative leading PFL frameworks.

## 2 PROBLEM STATEMENT

We focus on the standard personalized federated learning (PFL) problem, which aims to train a global model utilized as the initialization to produce the personalized model based on the local data of each client. In practical applications, each participating client can contain a local dataset with personal preferences and objectives. Limited by the privacy constraint, we cannot access the whole data distribution of clients to predetermined the global model targeted for optimality or adaptability to facilitate the local adaptation in each client. Thus, the major problem is to adaptively achieve the suitable trade-off between these two objectives of the global model, thus maintaining the solid personalized local models under non-IID data scenarios with different degrees of diversity among clients.

Basically, there are $C$ clients with the whole data distribution $(\mathcal{X}, \mathcal{Y})$ in the decentralized system. For an individual device with index $j \in C$, we assume the device-specific task objective $\Gamma^j$ is sampled from a task objective distribution $p(\Gamma)$, which follows the definition of the meta-learning Triantafillou et al. (2019). The local dataset $M^j$ of this client contains the train dataset $D^j : \left\{ (\mathbf{x}, \boldsymbol{y})^j \right\}$ and the test dataset $D'^j : \left\{ (\mathbf{x}', \boldsymbol{y}')^j \right\}$ that are disjoint. Both of them are sampled from the client-specific data distribution $D^j, D'^j \sim (\mathcal{X}, \mathcal{Y})^j \subset (\mathcal{X}, \mathcal{Y})$ and have $N^j$ and $N'^j$ samples, respectively. Besides, PFL aims to train a shared global model $\theta$ that is regarded as the initialization for producing high-quality personalized models $\phi^1, ..., \phi^C$.

*Non-iid data with diversity quantity $\zeta$.* The basic non-IID scenario utilized in our work is the distribution-based label non-IID data setting. The distribution of classes within one client follows the

Dirichlet distribution with parameter $0.5$. We define a mutually exclusive set in which each client augments the samples and randomly relabels the assigned classes. Thus, any two clients in this set potentially have different learning objectives and data distribution. Then, let $\zeta$ denote the proportion of non-mutually-exclusive devices in all $C$ devices. Thus, the diversity of the non-IID data can be quantified as $1 - \zeta$. $\zeta = 1$ means that local datasets of participating clients are subsets of the whole data distribution $(\mathcal{X}, \mathcal{Y})$, which defines a general non-IID data scenario in the FL.

Then, the $\theta$ is trained to get good personalized models by using $\min_\theta L(\theta) := \frac{1}{C} \sum_j L^j \left(\phi^j\right), \phi^j := \theta - \eta_l \nabla L^j(\theta)$ where $\eta_l$ is the update rate and $L^j : R^d \to R$ is the loss function of $M^j$ in device $j$. Thus, in local adaptation, the quality of $\phi^j$ depends on $\theta$ and $M^j$. On the one hand, when $\zeta = 1$, the learning of $\phi^j$ benefits from training $\theta$ to get optimality on the large global dataset $\left\{M^j\right\}_{j=1}^C$. On the other hand, a $\theta$ with high adaptability can be easily adapted to the local data in each client to obtain better $\phi^j$ when $\zeta = 0$.

The object of our generalizable PFL is to obtain high-quality personalized models efficiently under non-IID data scenarios with any diversity $\zeta \in [0, 1]$. We argue that adaptive local adaptation induces a suitable trade-off between the optimality and adaptability of the global model, which contributes to generalizable PFL. Thus, our goal is to achieve generalizable PFL by implementing adaptive local adaptation.

## 3 METHODOLOGY

In this section, we first present the issue of solely targeting the adaptability or optimality of the global model in leading works. Then, we provide insights into this issue through experiments and mathematical analysis. Based on the findings, we prove a theorem referred to as the adaptive trade-off theorem, which motivates the design of the adaptive federated meta-learning (AFML) framework.

### 3.1 ANALYSIS THE ISSUE FROM EXPERIMENTS

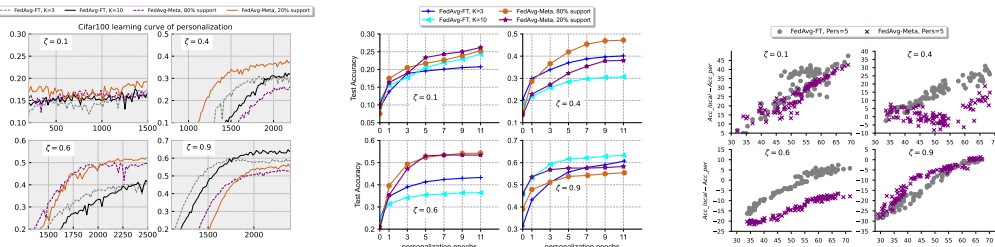

(a) Accuracy curve for two federated models under different $\zeta$.

(b) Personalized test accuracy under different personalization steps and $\zeta$.

(c) Accuracy of two federated models vs. local, trained-from-scratch models under different $\zeta$.

Figure 1: Illustration of the issue in the FedAvg-FT and FedAvg-Meta by performing experiments under non-IID data scenarios with different degrees of diversity $\zeta = 0.1, 0.4, 0.6, 0.9$ among clients. The E and Pers denote the local update steps and personalization steps, respectively. $s\%$ support in FedAvg-Meta is the proportion of support set used for training.

Our conducted experiments of the CIFAR100 dataset shown in Fig. 1 present the unstable performance of optimality-oriented with fine-tuning method FedAvg-FT Jiang et al. (2019) and adaptability-oriented FedAvg-Meta Chen et al. (2019) method in terms of the averaged personalized accuracy of clients and the learning efficiency. The local models were trained locally on clients' own data for 200 epochs with a learning rate of $0.005$.

Both two methods fail when there is a high diversity among clients, i.e. $\zeta = 0.1$. The upper left subfigure of Eq. 1 (a), (b) presents that they cannot converge while Eq. 1 (c) further shows that they obtain a bad personalized accuracy as compared with the model trained locally.

*Toward optimality*. FedAvg-FT obtains low accuracy and training speed in the medium diversity conditions, as shown by Fig. 1 $\zeta = 0.4, 0.6$. This illustrates that the initial model trained to be optimal in the whole data distribution is hard to adapt to the local dataset. For example, Fig. 1(b) shows that the corresponding accuracy of the personalized model improves less than $0.1$ than the initial one. But when there is low diversity $\zeta = 0.9$, FedAvg-FT outperforms the adaptability-oriented method with a large margin in all metrics. Especially in Fig Fig. 1(b), personalization of the FedAvg-FT can start from a powerful initial model to reach the accuracy of $0.6$ in only seven personalization epoches.

*Toward adaptability*. FedAvg-Meta can quickly converge to the best accuracy when the diversity is relatively high. Fig. 1(b) shows that the initial model with high adaptability can be trained with the local dataset to have a $0.3$ improvement in accuracy within 7 personalization steps. Otherwise, the personalization of FedAvg-Meta presents an opposite trend when the diversity is low $\zeta = 0.9$. It is trained by $1500$ communication rounds to obtain an unstable convergence while requiring 11 epochs to update the initial model with low performance to reach a competitive accuracy.

Fig. 1(c) further shows that both two methods are less accurate than the local models for the majority of participants when the diversity is high $\zeta = 0.1, 0.4, 0.6$. But consistent with the above analysis, the FedAvg-Meta is significantly better than the FedAvg-FT. This means that most clients can update the initial model with high adaptability to obtain a high-quality personal model. However, starting from a powerful initial model is preferable when $\zeta = 0.9$ because it is hard to enhance a weak initial model based on the small-size local dataset.

Both methods is obviously not conducive to generalizable PFL. When the diversity among clients is high, targeting the global model to perform well on the whole data distribution (i.e., optimality) makes it hard to be adapted to the local dataset. On the contrary, even though the global model can be highly adaptable, updating a weak model with respect to a small-scale local dataset produces a poor personalized model and damages the efficiency. As pointed by experimental results, solely targeting optimality (FedAvg-FT) or adaptability (FedAvg-Meta) of the global model leads to an unstable performance in non-IID data scenarios with different diversities. Therefore, this motivates us to propose the idea of adaptive local adaptation in which the global model is to train to obtain a suitable trade-off between optimality and adaptability based on the non-IID data condition.

## 3.2 INSIGHTS THROUGH THE INFORMATION THEORY

In the local adaptation of PFL, the initial model is updated based on the local data. Thus, from the information theory perspective, the information for learning the personalized model derives from the initial model and the local dataset. This makes us give insights into the discussed issue through the information flow in the local adaptation.

When $\zeta \to 1$, FedAvg-Meta with the meta-learning framework, tends to produce a complex global model with parameters $\theta$, which will only overfit datasets in part of clients containing similar task objective $\Gamma$. This phenomenon is defined as task overfitting in the meta-learning research Yin et al. (2020). Thus, in the local adaptation, $\theta$ effectively performs the test dataset without utilizing local training samples. This can be denoted as $q(\overline{y}'|\boldsymbol{x}', \phi)q(\phi|D, \theta) = q(\overline{y}'|\boldsymbol{x}', \phi)q(\phi|\theta)$, in which the local model $\phi$ is independent of the dataset $D$ in the client, such that $q(\overline{y}'|\boldsymbol{x}', \theta, D) = q(\overline{y}'|\boldsymbol{x}', \theta)$ where $\overline{y}'$ is the predicted label. Then, the participanting clients is unable to adjust the initial model based on personal local dataset.

In contrast, when there is high diversity $\zeta \to 0$ among clients, solely targeting the optimality in FedAvg-FT makes weights of the global model provide sparse prior information for personalized learning. The main reason is that the local data distribution deviates a lot from the global distribution. For instance, in the extreme case, the shared global model can be regarded as randomly initialized weights for each client to train on their own small-size local dataset. Thus, the training for a personalized model heavily relies on the information of the local dataset. This can be denoted as $q(\overline{y}'|\boldsymbol{x}', \phi, \theta)q(\phi|D, \theta) = q(\overline{y}'|\boldsymbol{x}', \phi)q(\phi|D)$, in which weights $\phi$ of the local model only dependent on the local dataset $D$. Personalized learning is equivalent to training random weights based on $D$, which damages the training speed and the performance.

An unsuitable trade-off hinders the generalizable PFL by causing the over-dependence on the global model or the local dataset in local adaptation. Therefore, we argue that the generalizable PFL can be

achieved by implementing the adaptive local adaptation that is formulated to balance the contributions of these two information sources in personalization.

### 3.3 Adaptive local adaptation via balancing information flows

The Bayesian inference shown by the graphical model in Fig. 2 *TRAIN* part presents the prediction process of the personalized local model $\phi^j$. The initial model $\theta$ is updated with the local dataset $D^j$ to generate the $\phi^j$, and then the $\boldsymbol{x}'$ is classified by the $\phi^j$. Thus, predicted label $\overline{\boldsymbol{y}}'$ is is obtained based on information derived from training data $D$ (i.e., $D \to \phi^j \to \overline{\boldsymbol{y}}'$) and $\theta$ (i.e., $\theta \to \overline{\boldsymbol{y}}'$). As discussed in Section 3.2, targeting the optimality and adaptability of the global model corresponds to over-depending on the former and latter information path in the local adaptation, respectively. These, we can achieve the adaptive local adaptation by balancing these two information flows.

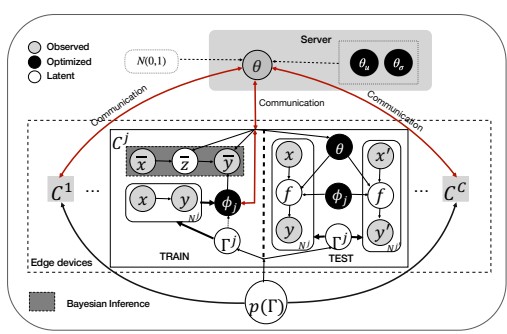

Figure 2: The graphical model that expresses the learning process in the personalized FL and our proposed method.

Motivated by the work in Yin et al. (2020) that utilizes the mutual information to express the relationship between variables, we naturally formulate the dependencies on two information flows as the mutual information $I(\phi; \theta)$ and $I(\phi; D)$. Therefore, the optimal information flow balance can be obtained by solving a constrained optimization problem, shown in Eq. 1. This problem aims to prevent the learning process from solely relying on one single information flow.

$$\max I(\overline{\boldsymbol{y}}'; \overline{\boldsymbol{z}}'|\theta, D)$$
$$s.t. I(\overline{\boldsymbol{y}}', D; \theta|\boldsymbol{x}') \leq I_c \tag{1}$$

where $I_c$ is the constant information constraint and $\overline{\boldsymbol{z}}'$ is a intermediate variable that is the hidden representation of the input $\boldsymbol{x}'$.

Given $\theta$ and $D$, the information flow path $\theta \to \overline{\boldsymbol{y}}'$ is motivated by maximizing mutual information between the target $\overline{\boldsymbol{y}}'$ and the intermediate variable $\overline{z}$ that is generated by $\theta$. This encourages to train a powerful global model that performs well on the global distribution. However, given the input $\boldsymbol{x}'$, we limit the dependence of local learning on $\theta$ by restricting the mutual information between the prediction and $\theta$ in the reasoning process. This mechanism encourages the initial model not to excessively store information of device-specific data in its parameter $\theta$, thereby increasing the importance of the local dataset in learning. Such constrain induces the global model with high adaptability.

### 3.4 An Adaptive Trade-off Theorem

Here, we solve the optimization problem in Eq . 1 by introducing a Lagrange multiplier $\beta$. Then, the problem is transformed into maximizing the formula, $\triangle = I(\overline{\boldsymbol{y}}'; \overline{\boldsymbol{z}}, |\theta, D) - \beta I(\overline{\boldsymbol{y}}', D; \theta|\boldsymbol{x}')$. Then, by modeling the $\theta$ as the stochastic variable following a Gaussian distribution, the variational inference is utilized to get the solution. After the mathematical derivation shown in Appendix A, we can determine a lower bound of this formula as:

$$\begin{aligned} \triangle \geq {} & E_{\boldsymbol{x}', \overline{\boldsymbol{y}}'} E_{\epsilon \sim N(0,I)} \left[ \log q(\overline{\boldsymbol{y}}'|\boldsymbol{x}', \theta, \epsilon) \right] \\ & - E \left[ KL(p(\overline{z}|\boldsymbol{x}', \theta)||r(\overline{z})) \right] - \beta E \left[ KL(p(\theta|D, \boldsymbol{x}', \overline{\boldsymbol{y}}')||r(\theta)) \right] \\ & + I(\boldsymbol{x}'; \overline{\boldsymbol{y}}'|\theta) - I(\overline{\boldsymbol{y}}'; D|\theta) \end{aligned} \tag{2}$$

where $r(\theta) \sim N(0, I)$ is a variational approximation to the target distribution $\theta$. In the first term of the equation, we present the detailed operation of the reparameterization trick while we ignore this detail in the following terms.

If the $I(\boldsymbol{x}';\overline{\boldsymbol{y}}'|\theta)$ is 0, the model predictions do not depend on the given $\boldsymbol{x}'$, leading to low accuracy. Thus, maximizing $I(\boldsymbol{x}';\overline{\boldsymbol{y}}'|\theta)$ term is equivalent to minimize training losses for high accuracy. Similarly, the high accuracy across clients leads to model predictions that are less dependent on local training data sets, making a smaller $I(\overline{\boldsymbol{y}}';D|\theta)$. Therefore, we conclude that maximizing the lower bound in Eq . 2 can be replaced by maximizing training accuracy and three regularizers shown by the theorem below.

**Theorem 1** (Adaptive trade-off theorem). *In non-IID data scenarios with different diversity, a suitable trade-off for learning high-quality personalization models is obtained by targeting the following equation during the training process.*

$$
\begin{aligned}
\max_{\theta} \; & E_{\boldsymbol{x}'} E_{\epsilon \sim N(0,I)} \left[ \log q(\overline{\boldsymbol{y}}'|\boldsymbol{x}',\theta,\epsilon) \right] - E\left[ KL(p(\overline{z}|\boldsymbol{x}',\theta)||r(\overline{z})) \right] \\
& - \beta E\left[ KL(p(\theta|D,\boldsymbol{x}',\overline{\boldsymbol{y}}')||r(\theta)) - E_{(\boldsymbol{x},y)\sim p^j(D)} \left[ L^j(\theta;\boldsymbol{x},y) \right] \right. \\
& - E_{(\boldsymbol{x}',y')\sim p^j(D'))} \left[ L^j(\phi^j;\boldsymbol{x}',y') \right]
\end{aligned}
\tag{3}
$$

*where $\theta$ represents the parameters of the global model trained on local data of decentralized clients $q(\theta|\left\{M^j\right\}_{j=1}^{C})$ and $\phi^j \sim q(\phi^j|D^j,\theta)$.*

The above theorem demonstrates that the adaptive local adaptation can be implemented by maximizing a novel objective function in training. Then, the learning system is able to find a suitable trade-off to produce high-quality personalized models in non-iid data scenarios with any degree of diversity.

## 3.5 Adaptive Meta-Learning algorithm

As pointed by federate meta-learning methods Chen et al. (2019); Khodak et al. (2019); Fallah et al. (2020), clients in the FL paradigm can be regarded as tasks in the meta-learning framework. Then, we can have the support set $D$, and the query set $D'$. Based on these definitions, the last two terms in Eq 3 of Theorem 1 are naturally general training objectives of meta-learning. The initial model FF is updated using the support set to minimize the loss in the query set. Therefore, based on Theorem 1, we propose the *adaptive meta-learning framework* (AFML) to achieve generalizable PFL with adaptive local adaptation. The training schema of AFML naturally follows the meta-learning framework Finn et al. (2017).

Theorem 1 also introduces three additional terms, which can be expressed as three trade-off regularization terms. Thus, the regularizer $L_{TOFF}$ is formulated as follows.

$$
\begin{aligned}
L_{TOFF} = & \frac{\alpha}{N'} \sum_{\boldsymbol{x}',y' \in D'} E_{\epsilon \sim p(\epsilon)} \left[ -\log q(\overline{y}'=y'|\theta,\boldsymbol{x}',\epsilon) \right] && L_{ENC} \\
& + \frac{\alpha}{N'} \sum_{\boldsymbol{x}',y' \in D'} KL\left(p(\overline{z}|\boldsymbol{x}',\theta)||r(\overline{z})\right) && L_{CONS} \\
& + \beta KL(p(\theta|D,D')||r(\theta)) && L_{CMPX}
\end{aligned}
\tag{4}
$$

where the $L_{ENC}$ encourages an encoding that is maximally informative about the target. $L_{CONS}$ gives constraints to the latent representation of $\overline{z}$, which is similar to the work Alemi et al. (2016). $L_{CMPX}$ motivates an initial model with low complexity.

For the training process based on all participating client datasets, the third item $L_{CMPX}$ can be further written as $E\left[ KL(p(\theta|M)||r(\theta)) \right]$. Therefore, when we set $\theta \sim N(\tau)$ with $\tau = (\theta_u,\theta_\sigma)$, the $L_{CMPX}$ can be put in the server and computed as $L^j_{CMPX} = KL(p(\theta|\theta_u,\theta_\sigma)||r(\theta))$.

Finally, to achieve the *Adaptive trade-off theorem*, we design split the global model into two modules, including the encoding network $f$ and the classification network $h$. The encoding network is trained to learn the hidden representation of the observation $\boldsymbol{x}$, while $h$ is used to make the prediction based on $\overline{z}$. One intuition of such design is to alleviate the inconsistent between different local datasets by learning a compact shared space. Then, we add the regularization on weights $\theta$ of the encoding network and leave the weights of $h$ unrestricted.

The corresponding training algorithm is shown by *Algorithm 1*.

---

**Algorithm 1:** Learning algorithm of AFML

---

**input** : OuterLoop/Server learning rate $\eta_1, \eta_2$; InnerLoop/Client learning rate $\eta$;
Regularization coefficient $\beta$; #C participating clients

1 **Initialization**: Initialize weights distribution $\theta \sim N(\tau)$ with $\tau = (\theta_u, \theta_\sigma)$ for the encoding network $f$; Initialize weights $\widehat{\theta}$ for classification network $h$. Initialize local models $\phi = \left\{ \phi^1, ..., \phi^j, ..., \phi^C \right\}$;

2 **for** *communication round* $t = 1, 2, ...$ **do**

3      Randomly select $K$ clients $C_t$ from the total $C$ clients;

4      Sample $\theta_t$ from $N(\tau)$ with reparameterization;

5      Distribute $\theta_t$ and $\widehat{\theta}$ to selected clients;

6      **for** *each client $j \in C_t$ in parallel* **do**

7          $g_c^j, g_g^j \leftarrow clientUpdate\left( j, \theta_t, \widehat{\theta}_t \right)$;

8      **end**

9      $\widehat{\theta} = \widehat{\theta} - \frac{\eta_1}{K} \sum_{j \in C_t} g_c^j$;      $\triangleright$ Aggregating the updates for the global classification network

10     $\tau = \tau - \frac{\eta_2}{K} \sum_{j \in C_t} g_g^j$;      $\triangleright$ Aggregating the updates for the global encoding network

11 **end**

12 **Function** `ClientUpdate`$(j, \theta, \widehat{\theta})$**:**

13      $M^j \leftarrow \left( D^j = \left( \mathbf{x}^j, \boldsymbol{y}^j \right), D'^j = \left( \mathbf{x}'^j, \boldsymbol{y}'^j \right) \right)$;

14      Encode observation to the hidden feature $\mathbf{z}^j = f(\mathbf{x}^j; \theta)$, $\mathbf{z}'^j = f(\boldsymbol{y}'^j; \theta)$;

15      Update the initial model, $\phi^j \leftarrow \widehat{\theta} - \eta \nabla_{\widehat{\theta}} L_{D^j}^{\widehat{\theta}}$, $L_{D^j}^{\widehat{\theta}} = \frac{1}{Z} \sum_{\mathbf{z}^j, \boldsymbol{y}^j \in D^j} \ell \left( h \left( \mathbf{z}^j; \widehat{\theta} \right), \boldsymbol{y}^j \right)$;

16      Obtain the loss on the query set, $L_{D'^j}^{\phi^j} = \frac{1}{Z} \sum_{D'^j} \ell \left( h \left( \mathbf{z}'^j; \phi^j \right), \boldsymbol{y}'^j \right)$;

17      Compute the gradients for the classification network, $g_c^j = \nabla_{\widehat{\theta}} L_{D'^j}^{\phi^j}$;

18      Compute the gradients for the global model, $g_g^j = \nabla_\tau \left( L_{D'^j}^{\phi^j} + \alpha L_{TOFF}^j \right)$ where $L_{TOFF}^j = L_{ENC}^j(D', \theta_t) + L_{CONS}^j(D', \theta_t) + \beta L_{CMPX}^j(\theta_t)$;

19      **return** $g_c^j, g_g^j$; ;

---

## 4    RELATED WORK

Federated learning (FL) Zhao et al. (2018); Kairouz et al. (2019); Li et al. (2020) is a machine learning paradigm in which devices collaboratively train a global model while keeping the training data decentralized. With conventional federated learning (FL) McMahan et al. (2017), the global model is only trained to develop a common output for all devices. However, the inconsistent learning objectives and data heterogeneity among devices in non-iid scenarios introduce a new personalization requirement Chen et al. (2019). Thus, we have witnessed significant progress in works Jiang et al. (2019); Fallah et al. (2020); Mansour et al. (2020); Yu et al. (2020); Khodak et al. (2019) of personalized federated learning (PFL).

Due to inherent diversity among local data shards and objectives, many works Chen et al. (2019); Jiang et al. (2019); Fallah et al. (2020) designed target functions that maximum the adaptability of the global model. Especially, Chen et al. (2019) firstly noticed that such a target function could be achieved by introducing the meta-learning framework to PFL.

However, it is still challenging for existing methods to maintain performance when there are non-iid scenarios with different degrees of diversity among devices. Arivazhagan et al. (2019) discussed that sole pursuing adaptability damages the personalization when devices require a powerful initial model in local adaptation. They solved this problem by training a base layer to be optimal and changing only the last layer to personalize in individual devices.

The latest works proposed to maintain personalization performance in non-iid scenarios with various diversities were Deng et al. (2020); Hanzely & Richtárik (2020); Khodak et al. (2019); Mansour et al. (2020). Khodak et al. (2019) specifically pointed out that existing PFL methods' performance is unstable when diversity among different devices' data changes. These works alleviated the issue by seeking joint optimization of the global model and personalized models. The proposed methods, such as Deng et al. (2020) heavily relied on trainable mixing weights that are updated during the learning process.

## 5 EXPERIMENTS

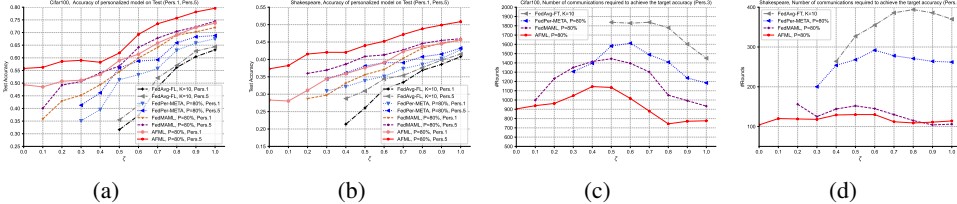

| (a) | (b) | (c) | (d) |

Figure 3: Test accuracy of personalized models (a) (b) and Communication rounds (c) (d) required to reach to target accuracy under $\zeta \in [0, 1]$. Our AFML is compared with FedAvg-FT, FedAvg-Meta, and FedMAML based on the CIFAR100 (a) (c) and Shakespeare (b) (d) datasets.

In this section, we evaluate the performance of the proposed AFML on CIFAR-100 and Shakespeare Caldas et al. (2018) datasets under the basic non-IID setting described in the Section 2 with the diversity $\zeta \in [0, 1]$. Then, following the three metrics described in the Section 3 of the Appendix, AFML is compared with three state-of-the-art methods, including FedAvg-FT Jiang et al. (2019), FedPer-Meta Fallah et al. (2020), and FedMAML Deng et al. (2020) where FedPer-Meta is an enhancement work of FedAvg-Meta Chen et al. (2019). Limited by the space, more settings and experimental results are described in Section 3 of the appendix.

### 5.1 RESULTS

**Accuracy performance.** Fig 3 presents that AFML achieves highest accuracy compared to three leading frameworks under non-IID scenarios with any degrees of diversity, yet with minimum communication rounds used.

In the non-IID scenario with highly diversity $\zeta \leq 0.3$, the average personalization accuracy of AFML is around 56% and 39% at CIFAR-100 and Shakespeare, respectively. However, other frameworks cannot converge. In the non-IID scenario with low diversity $\zeta \geq 0.5$, AFML has a significant performance improvement and gets the highest accuracy, which is 5.25% higher than FedMAML, 9.79% higher than FedPer-Meta and 18.69% higher than FedAvg-FT on CIFAR100. AFML also performs well on the Shakespeare dataset. Moreover, compared with other methods, after performing five personalization epochs (Pers.5 Acc), the average personalization accuracy of AFML is 5.2% and 7.4% higher than its Pers.1 Acc on CIFAR-100 and Shakespeare, respectively.

The main reason is that AFML not only provides a strong point for personalization but also improves the adaptability of the global model. The insight observation from our theorem is that AFML trains an optimal global model with low generalization error while also targets adaptability. And it can achieve a suitable trade-off under non-iid scenarios with different diversity. This also contributes to the high convergence speed.

**Efficiency performance.** As shown in Fig. 4, compared with other algorithms, the computation cost of AFML achieves a significant reduction. In particularly, the Bytes of AFML is respectively 22.5%, 55.7%, and 67.3% less than FedMAML, FedPer-Meta and FedAvg-FT in average on both datasets.

**Fairness Comparison.** The results presented in Fig. 4(b) and Fig. 5 demonstrates that AFML has the ability to learn high-quality personalized models for majority clients. For both two datasets, AFML leads to more clients with higher accuracy compared to the model trained locally. Moreover,

our proposed AFML does not encourage the sacrifice of fairness to obtain higher average accuracy, reflecting the effectiveness of trade-off regularization terms. The main reason is that each device can optimize its information flow to obtain a suitable trade-off for better personalization.

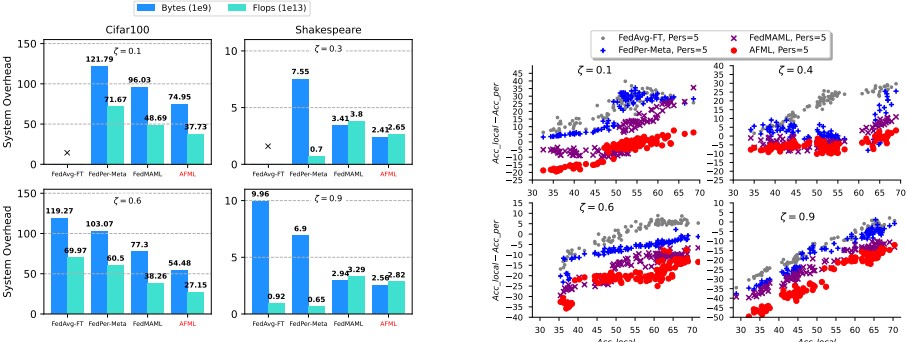

(a) System overhead for achieving a target accuracy in different methods

(b) Accuracy of four models vs. local, trained-from-scratch models under different $\zeta$

Figure 4: Comparison of our AFML with three leading PFL methods on CIFAR100 and Shakespeare datasets. For System overhead in (a), we set the target accuracy as its test accuracy when convergence. (The symbol "×" represents that the algorithm cannot converge.)

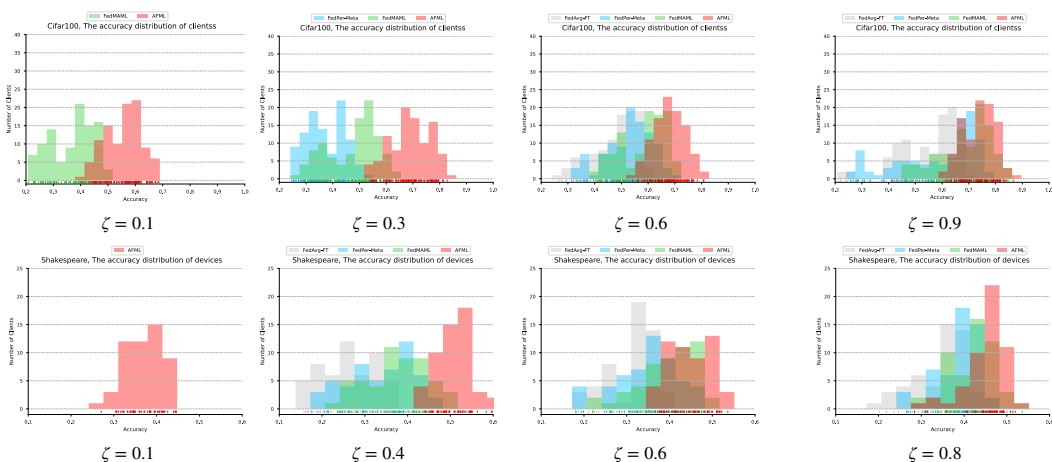

Figure 5: Accuracy distribution of devices in AFML compared to FedAvg-FT, FedPer-Meta, and FedMAML under different non-IID settings. The first row presents the results of the CIFAR100, while the second row presents the Shakespeare.

## 6 CONCLUSION

We have pointed out the connection between the generalizable PFL and learning objectives (i.e., optimality and adaptability) trade-off of the global model. An in-depth study through experiments and mathematical analysis shows that a suitable trade-off can be obtained by implementing adaptive local adaptation. A proven theorem referred to as the *adaptive trade-off theorem* further shows balancing information flow in local adaptation contributes to the desired suitable trade-off. These theoretical insights guide our design of a novel adaptive federated meta-learning(AFML) framework that achieves generalizable PFL with adaptive local adaptation. Through the empirical experiments on non-iid data scenarios with an extensive diversity among devices, we have demonstrated that the proposed AFML is superior to alternative learning frameworks in terms of personalized learning quality and convergence speed.

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
