# OpenReview forum: "Towards Generalizable Personalized Federated Learning with Adaptive Local Adaptation"
_ICLR.cc/2022/Conference — ICLR 2022 Submitted_

### Official Review · Reviewer_Mvwu · 2021-10-27

**Correctness:** 4
**Technical Novelty And Significance:** 2
**Empirical Novelty And Significance:** 2
**Recommendation:** 5
**Confidence:** 3

**Main Review:**

The idea of using information theory to motivate appropriate regularization in meta-learning seems interesting. However, this contribution is mainly made in [1]. The paper borrows the techniques from [1] to solve personalized federated learning (PFL) tasks. Unfortunately, although the authors have put much effort in explaining their algorithms, I still have little insight about the motivation of algorithm design and derivation of the regularization term. At least to me, the main techniques used in this paper seem very similar to [1]. I suggest the authors to explain what is the key difference between your paper and [1], and what are challenges of applying [1] to PFL problem and how did you solve them.

Another main concern is about tuning the parameters in your proposed algorithms. I don’t see any suggestions on how to choose tuning parameters such as $\alpha$ and $\beta$. Thus, I have doubts that the superiority of experimental performance may come from flexible tuning parameter space. I would suggest the authors to carefully design experiments to test the robustness of parameter choice or at least explain how you choose hyperparameters of your algorithms and competitors.

In addition, the writing is not in good shape. I observed a lot of typos and grammar mistakes through the paper. I pointed two places below, but there are much more errors. Please check them carefully. And the figures shown in the paper are too small to read. Besides, although the authors explained the information theory motivation, I still have little insight. I think one important ingredient is using generative model to learn some latent representation. However, the authors did not carefully explain that. I would suggest the authors to explain your algorithm design step by step so that people who are not familiar with relevant works can also follow. The high-level intuition can come after or accompany the detailed algorithm description.

Minor issue:
1. Grammar mistakes and typos. For example, in page 1, “Then, in the local adaptation Yu et al. (2020), it trains …” should be “Then, in the local adaptation, Yu et al. (2020) trains ….”. Citation should be treated as a subject. Please check similar problems in the paper. And in page 4, “Both methods is obviously not conducive to generalizable PFL” should be “Both methods are obviously not conducive to generalizable PFL”. There are many grammar mistakes. Please have a check on writing.
2. Since the whole paper talks about diversity level, I think it will be better to sue $\zeta$ instead of $1-\zeta$ to refer to it, as I feel it more natural to think about a larger $\zeta$ indicates a larger diversity level rather than the opposite.

[1] Yin M, Tucker G, Zhou M, et al. Meta-learning without memorization[C]//International Conference on Learning Representations. 2020.

**Summary Of The Paper:**

The paper frames the personalized federated learning (PFL) problem as a meta-learning task. In order to achieve the appropriate trade-off between global optimization and local adaptability, the authors utilize the recent work [1] to add a meta-learning regularization term for better personalization effect. The paper then demonstrates the benefits of their proposed method over the other alternatives by implementing extensive experiments.

[1] Yin M, Tucker G, Zhou M, et al. Meta-learning without memorization[C]//International Conference on Learning Representations. 2020.

**Summary Of The Review:**

Given the main techniques is borrow from existing literature, I have concerns about original contribution. Besides, the writing is not in good shape and needs major revision. I would suggest the paper for further improvement rather than accepting it at current state.

---

### Official Review · Reviewer_uetq · 2021-10-30

**Correctness:** 4
**Technical Novelty And Significance:** 3
**Empirical Novelty And Significance:** 3
**Recommendation:** 6
**Confidence:** 4

**Main Review:**

The proposed loss with regularizer, to combat memorization (also maximize the information flow) is an interesting and promising method. Apart from the positive parts, I have a few comments:

(1) The dataset with different zeta is good for synthetic dataset (which AFML adapts well with differently labeled dataset). However, FL typically works on real world dataset. Applying AFML on more real world dataset (e.g. EMNIST-62 without randomize the labels) could further strengthen the position of this paper.

(2) The exact differentiate factors between this paper and Yin et al could be made, to make this paper's contribution more easy to digest. I need to cross-read this paper and Yin et al multiple times to understand the differentiating factor.

(3) The hyperparameter alpha and beta in AFML is pretty critical for final performance. I see on appendix that it is fixed, a badly configured alpha and beta will not work. Some discussion on the loss hyperparameter could be helpful (e.g when will the performance on personalization hurts global model performance, there should be some trade-off, which is missing in discussion). The potential side effects of combating memorization (e.g hurt performance under some cases, since memorization could be very useful in many cases) is not shown in this paper.

**Summary Of The Paper:**

This paper is a Federated Learning (FL) extension version of "Meta Learning without Memorization" (Yin et al 2020). The author uses FL personalization loss, together with the three regularizers (information loss(L_ENC), activation regularization(L_CONS) and weight regularization(L_CMPX)). The activation and weight regularization are very close to Yin et al, while the information loss L_ENC is the difference, since Meta Learning aims only at fast adaptation, while FL personalization also aims to learn a good global model without adaptation: L_ENC could be pretty useful to learn a feature extraction network.

The proposed loss function and regularizer, show some improvement of personalization on non-IID dataset with different level of zeta, which indicates that panelizing memorization could improve personalization performance (test acc on average and individual improvement, and communication rounds). It is an interesting work interplaying between meta learning and FL.

**Summary Of The Review:**

Combating memorization is very important to improve FL personalization performance, the proposed method is promising.
However, some experiments on real-world experiments are missing to support the claim.
Moreover, the linkage between meta learning work (Yin et al) and the proposed work could be better addressed.
Finally, the negative effects of combating memorization is missing, which could further strengthen the paper.

Overall, I give score 6, and happy to conduct discussion with the authors and other reviewers.

---

### Official Review · Reviewer_qKrd · 2021-11-03

**Correctness:** 3
**Technical Novelty And Significance:** 2
**Empirical Novelty And Significance:** 2
**Recommendation:** 3
**Confidence:** 4

**Main Review:**

Contributions

1. Using mutual information to deal with the global learning vs personalization trade-off in federated learning is a novel approach and may yield promising results with further development.
2. The empirical results presented are somewhat promising, showing accuracy and efficiency improvements over FedAvg+FT and PerFedAvg with very limited details provided.

Major issues

1. The writing is very poor and makes the technique difficult to understand. The problem formulation is not clear, for example, it is not defined what the global dataset is or what the local loss functions are. The motivation for equation (1) is very vague, and intuition for the derivation of (2) and Theorem 1 are not clear. It is not clear what the algorithm is actually doing (e.g. what is the encoding network?). There are many grammatical mistakes.

2. Theorem 1 cannot be considered a theorem. "A suitable trade-off for learning personalized models" is not defined so the statement is meaningless.

3. Very few experimental details are provided. Granted not all experimental details need to be provided in the main body, but for a paper whose main contributions are empirical, there are far too few details given. It is not explained how accuracies are computed, how many clients are in the dataset, how many samples are in each client's datasets, how fine-tuning is implemented for FedAvg+FT, what the difference between FedAvg-Meta (known as PerFedAvg in the original paper) and FedMAML is, and how the hyperparameters for these methods are chosen. It is not explained why FedAvg+FT does not converge in some cases, or what not converging means (and the lack of convergence is strange -- I suspect this is a failure of initialization). Without these details it is difficult to glean the meaning of the results.

4. Given the breadth of personalized federated learning methods, it is insufficient to compare against only three baselines (especially since two of the baselines are very similar).

Minor issues

1. The writing in the plots is too small.

2. The claim of improved fairness due to the proposed method is not substantiated with a definition of fairness and a corresponding metric showing the improvement of the proposed method.

Comparison with related work

The authors discuss works that combine federated learning and meta-learning at a high level (Fallah et al. 2020, Chen et al. 2019, Jiang et al. 2019) and mention other personalized federated learning techniques without explanation. Also, they ignore works on personalized FL from 2021. The use of mutual information in federated learning is novel to the best of my knowledge, but the proposed approach draws inspiration from Yin et al. 2020. The authors mention this work but do not clearly explain how their technique differs.

Yin et al., META-LEARNING WITHOUT MEMORIZATION, ICLR 2020

**Summary Of The Paper:**

This paper poses an "optimality vs adaptability" tradeoff for federated learning, where optimality corresponds to strong performance on a global dataset and adaptability corresponds to a model's ability to adapt to perform well on local datasets. To deal with this tradeoff, this paper seems to draw motivation from the modified MAML approach of Yin et al. 2020 that is motivated by preventing the meta-parameters from memorizing the meta-training data. In the federated learning setting considered here, the current paper tries to balance the mutual information between the user-updated parameters and the global data ("optimality") and the user-updated parameters and the meta-parameters ("adaptability"). The proposed approach is similar to MAML in that the meta-parameters are a globally shared initialization for gradient descent, and the user-updated parameters are the result of gradient descent from the shared initialization (i.e. the meta-parameters). Some theoretical analysis and preliminary empirical results are provided.

**Summary Of The Review:**

Considering the above mentioned points, I believe that this paper is below the bar.

---

### Decision · Program_Chairs · 2022-01-20

**Decision:**

Reject

**Comment:**

This paper approaches personalized federated learning from the perspective of meta-learning and use the mutual information framework developed in a recent work to regularize local model training. All the reviewers consider the writing very poor and hard to understand, and the contributions not sufficient for acceptance.